# Impact of the scale-up of piped water on urogenital schistosomiasis infection in rural South Africa

**Frank Tanser**[1,2,3,4]*, **Daniel K Azongo**[5], **Alain Vandormael**[1,2,6], **Till Bärnighausen**[1,4,7,8], **Christopher Appleton**[9]

[1]Africa Health Research Institute, KwaZulu-Natal, South Africa; [2]School of Nursing and Public Health, University of KwaZulu-Natal, Durban, South Africa; [3]Centre for the AIDS Programme of Research in South Africa (CAPRISA), University of KwaZulu-Natal, Durban, South Africa; [4]Institute of Epidemiology and Health Care, University College London, London, United Kingdom; [5]Navrongo Health Research Centre, Ghana Health Service, Navrongo, Upper East Region, Ghana; [6]KwaZulu-Natal Research Innovation and Sequencing Platform (KRISP), College of Health Sciences, University of KwaZulu-Natal, Durban, South Africa; [7]Department of Global Health and Population, Harvard T.H. Chan School of Public Health, Boston, United States; [8]Institute for Public Health, University of Heidelberg, Heidelberg, Germany; [9]School of Life Sciences, University of KwaZulu-Natal, Durban, South Africa

**Abstract** Recent work has estimated that sub-Saharan Africa could lose US$3.5 billion of economic productivity every year as a result of schistosomiasis and soil-transmitted helminthiasis. One of the main interventions to control schistosomiasis is the provision of safe water to limit the contact with infected water bodies and break the cycle of transmission. To date, a rigorous quantification of the impact of safe water supplies on schistosomiasis is lacking. Using data from one of Africa's largest population-based cohorts, we establish the impact of the scale-up of piped water in a typical rural South African population over a seven-year time horizon. High coverage of piped water in the community decreased a child's risk of urogenital schistosomiasis infection eight-fold (adjusted odds ratio = 0.12, 95% CI 0.06–0.26, p<0.001). The provision of safe water could drive levels of urogenital schistosomiasis infection to low levels of endemicity in rural African settings.
DOI: https://doi.org/10.7554/eLife.33065.001

*For correspondence:
ftanser@gmail.com

**Competing interests:** The authors declare that no competing interests exist.

## Introduction

Schistosomiasis is a chronic parasitic disease caused by blood flukes of the genus *Schistosoma* and transmitted by snails found in fresh water bodies that have been contaminated by *Schistosoma* eggs. Approximately, 240 million people worldwide (**WHO, 2010**; **Stothard et al., 2009**) are infected with schistosomiasis, with a disease burden of 3.3 million disability-adjusted life-years (**Murray et al., 2012**). In sub-Saharan Africa alone, an estimated 163 million people were infected in 2012 (**Lai et al., 2015**). Following the London declaration (**Uniting to Combat Neglected Tropical Diseases, 2012**), the World Health Organization (WHO) has set a goal of increasing mass drug administration coverage to 75% of at-risk children in endemic countries (**WHO, 2013**). Despite progress in many so-called 'neglected tropical diseases', the fourth progress report of the London Declaration records that schistosomiasis has the lowest coverage of all helminth diseases treatable by mass drug administration (**Uniting to Combat Neglected Tropical Diseases, 2016**).

For achieving sustained control, elimination, or eradication of schistosomiasis, improvements of water, sanitation, and hygiene infrastructure and modification of risk behaviour are necessary (*Freeman et al., 2013*; *WHO, 2015*; *Steinmann et al., 2006*; *Echazú et al., 2015*). One important approach is the provision of safe water supplies to reduce the need for contact with contaminated water bodies and diminish the risk of schistosomiasis transmission. However, quality evidence for the impact of provision of safe water on schistosomiasis infection is currently lacking. Recently, a systematic review concluded that previous studies on this topic were of poor quality and likely confounded by multiple socio-economic and other variables (*Grimes et al., 2014*). The authors conclude that new research is needed 'on the relationships between water, sanitation, hygiene, human behaviour, and schistosome transmission'. In this regard, the previously used exposure measures of household or individual access to safe water (*Grimes et al., 2014*) will underestimate the true health impacts of the provision of safe water at a population level. This is because the introduction of safe water supplies into a community offers protection from schistosomiasis infection in two ways. Firstly, it offers *direct protection* to individuals with access to safe water by reducing their contact with infected water bodies through household domestic water collection activities. Secondly, the scale-up in access to safe water will confer *indirect protection* to members of a community in a manner that is analogous to the concept of 'herd immunity' (*Fine et al., 2011*). This form of protection is conferred through a reduction in the number of contacts that infected individuals have with open water bodies leading to a decrease in the overall levels of intensity of infection in the surrounding community. Thus, the greater the proportion of individuals in a community who are not actively contributing to the perpetuation of the schistosomiasis transmission cycle, the smaller the probability that those who interact with open water bodies will come into contact with free swimming *cercariae* released by infected snails.

Hence, to capture the true overall population-level impact of the introduction of piped water, it is necessary to move beyond a simple household/individual level exposure approach towards a sensitive community-level exposure estimate that quantifies the coverage of piped water in the surrounding local community over an extended period of time preceding the measurement of the disease outcome. To date, no studies have quantified the (likely non-linear) relationship between coverage of piped water (%) at the community level and corresponding reductions in risk of schistosomiasis infection. From an epidemic control and eradication perspective, quantification of such a relationship would provide the best indicator of the true overall impact of the scale-up in piped water and would reveal critical coverage targets for achieving meaningful reductions in risk of schistosomiasis infection.

To accomplish this goal, we nested a *Schistosoma haematobium* survey within one of Africa's largest population-based cohorts in which all children and their families (N ≈ 90,000) have been intensively followed up in an endemic area of rural KwaZulu-Natal, South Africa. We exploit the heterogeneity in the space-time scale-up of piped water over seven years and detailed longitudinal risk factor information to provide a rigorous quantification of the impact of the introduction of piped water on risk of urogenital schistosomiasis infection measured in children living in the study community. We then validate the veracity of our findings using an instrumental variable approach.

## Results

Of the 2105 participants tested in the parasitological survey, 353 were infected with *Schistosoma haematobium*. The overall prevalence was 16.8% (95% CI: 15.2–18.4) and the prevalence of heavy infection (≥50 eggs/10 ml of urine) was 9.5% (95% CI: 8.3–10.8). Prevalence was twice as high in boys compared to girls (22.6% versus 11.4%) and peaked at 30% in boys at age 14 years and 15% in girls at age 12 years (*Figure 1*). Prevalence of heavy infection was 13.5% in boys (and peaked at 21.4% at age 14 years) and 5.7% in girls (peaking at 8.7% at age 12 years). 57% of all infected participants had heavy infections with a geometric mean egg burden of 65 eggs/10 ml urine. Overall, 80% of the egg burden was borne by just 3.4% of all pupils tested (20.4% of all infected participants).

Overall, 67% of participants who were present at school on the day of testing had parental consent and assent and provided a urine specimen. Only 2.2% of linked participants reported receiving praziquantel in the last 12 months (*Table 1*). Participation in the parasitological survey did not differ significantly by piped water access or any of the environmental and socio-economic control variables used in the analysis. However, there were differences in levels of participation by school (p<0.05)

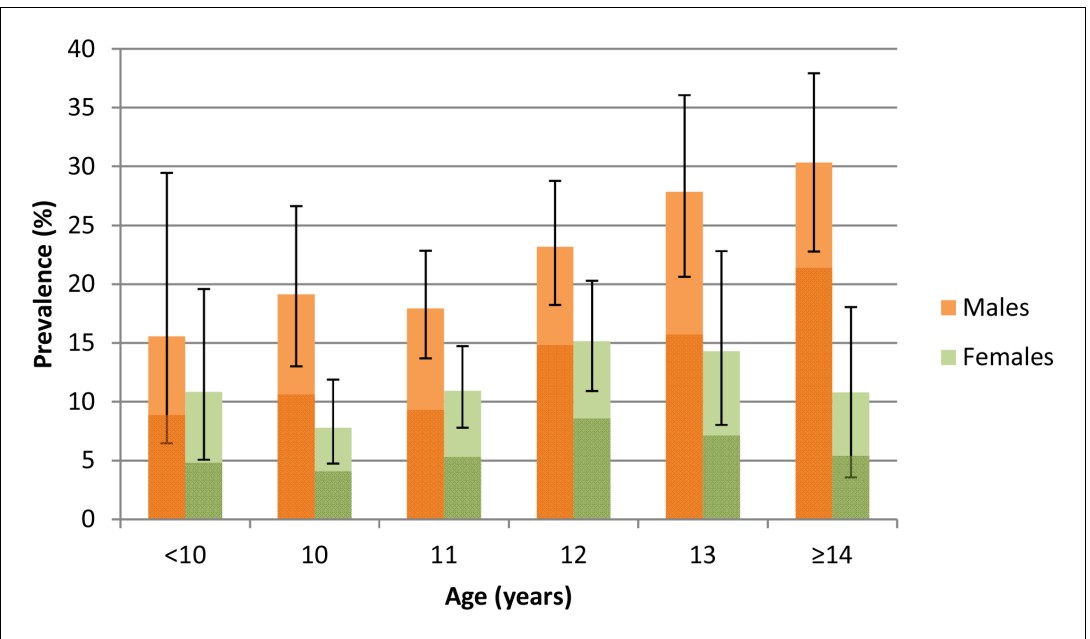

**Figure 1.** Prevalence (95% CI) of *Schistosoma haematobium* infection by age and sex among children taking part in the parasitological survey (N = 2,105). Darker colours represent heavy infections (≥50 eggs per 10 ml urine).
DOI: https://doi.org/10.7554/eLife.33065.002

and small differences by age and sex (p<0.1). After standardising for these differences according to age, sex and school, the prevalence remained virtually unchanged at 16.9%.

The geospatial analysis using the Gaussian kernel approach revealed substantial spatial variation in prevalence of urogenital schistosomiasis across the different communities within the surveillance area (*Figure 2*). The kernel-derived prevalence of urogenital schistosomiasis varied between 0% and 85% and revealed two clear regions of high transmission where prevalence was >50%. A formal micro-geographical clustering analysis using the Kulldorff spatial scan statistic revealed four significant foci of infection around open water collection sites (relative risk = 2.99–5.63, p<0.05; *Figure 2*, *Table 2*). Two clusters of infection were located along the Nsane River in the northeast of the surveillance area (clusters 2 and 3) and the other two located around water bodies towards the south of the study area.

Between 2001 and 2007, the provision of piped water in the surveillance area nearly doubled from 42% to 80% (*Figure 3*). The community-based mean coverage of piped water between 2001 and 2007 was 61.7% (*Figure 3*, bottom right). There is a high degree of temporal and spatial consistency in the patterns of expansion of piped water access across the five respective surveys, providing assurance of the quality of the underlying population-based data.

We identified a clear dose-response between the risk of urogenital schistosomiasis infection and piped water coverage in the surrounding local community (*Figure 4a*). Participants living in communities with a high coverage of piped water (median coverage = 95.5%) were eight times less likely to be infected (aOR = 0.12, 95% CI 0.06–0.26, p<0.001; Model 1 of *Table 3*) in comparison to those living in communities with lowest coverage (median coverage = 8.4%). This reduction was greater for girls than boys (aOR = 0.07 versus aOR = 0.17 in boys; Model 1 of *Table 4*) after controlling for key socio-economic and environmental predictors of infection.

Using a continuous measure of community-level piped water coverage, we found that every 1% increase in the coverage of community piped water was independently associated with a 2.5% decrease in the odds of infection (aOR = 0.975, 95% CI 0.966–0.985, p<0.001; see *Figure 4b*). The relationship is a non-linear one because it is derived from the piped water coverage regression coefficient (–0.025) and then exponentiated to derive an odds ratio. The resulting relationship would imply that relatively modest levels of piped water coverage in a given community could have a large impact on reduction in the odds of infection. For example, a piped water coverage of 30% would

**Table 1.** Characteristics of the primary school children within the survey who were linked to the population-based cohort (N=1976).

| Covariate | Total | Infected (%) | (95% CI) |
|---|---|---|---|
| **Gender** | | | |
| Female | 1016 | 117 (11.5) | (7.7–16.9) |
| Male | 960 | 217 (22.6) | (17.5–28.7) |
| **Age group** | | | |
| 9 | 118 | 14 (11.9) | (7.1–19.2) |
| 10 | 366 | 44 (12.0) | (6.8–20.3) |
| 11 | 592 | 87 (14.7) | (11.3–18.9) |
| 12 | 469 | 92 (19.6) | (13.6–27.4) |
| 13 | 222 | 49 (22.1) | (15.8–30.0) |
| ≥14 | 209 | 48 (23.0) | (15.1–33.3) |
| **Community piped water (quintiles)\*** | | | |
| (Lowest) 1 | 399 | 102 (25.6) | (13.0–44.1) |
| 2 | 415 | 87 (21.0) | (15.7–27.4) |
| 3 | 397 | 65 (16.4) | (10.5–24.6) |
| 4 | 386 | 33 (8.5) | (5.4–13.2) |
| 5 | 379 | 47 (12.4) | (8.9–16.9) |
| **Household access to water** | | | |
| No piped water | 304 | 53 (17.4) | (9.9–28.8) |
| Piped water | 1672 | 281 (16.8) | (12.9–21.6) |
| **Household assets quintiles** | | | |
| (Poorest) 1 | 398 | 83 (20.9) | (13.3–31.1) |
| 2 | 387 | 63 (16.3) | (11.7–22.1) |
| 3 | 389 | 63 (16.2) | (11.7–22.0) |
| 4 | 365 | 56 (15.3) | (11.5–20.1) |
| 5 | 359 | 57 (15.9) | (10.4–23.6) |
| Missing | 78 | 12 (15.4) | (9.5–23.9) |
| **School grade** | | | |
| Grade 5 | 1039 | 186 (17.9) | (12.2–25.5) |
| Grade 6 | 937 | 148 (15.8) | (10.4–23.2) |
| **Praziquantel in the last 12 months** | | | |
| No | 1933 | 321 (16.6) | (12.5–21.8) |
| Yes | 43 | 13 (30.2) | (17.3–47.3) |
| **Altitude** (meters above sea level) | | | |
| <50 | 76 | 29 (38.2) | (19.3–61.4) |
| 50–100 | 641 | 168 (26.2) | (19.2–34.7) |
| 100–150 | 875 | 108 (12.3) | (9.5–16.0) |
| 150–200 | 296 | 22 (7.4) | (4.4–12.4) |
| >200 | 88 | 7 (8.0) | (3.5–17.2) |
| **Distance to water body** | | | |
| <1 km | 606 | 112 (18.5) | (14.2–23.7) |
| 1–2 km | 618 | 122 (19.7) | (14.0–27.0) |
| 2–3 km | 376 | 68 (18.1) | (10.8–28.7) |
| >3 km | 376 | 32 (8.5) | (5.4–13.2) |
| **Toilet in household** | | | |

*Table 1 continued on next page*

*Table 1 continued*

| Covariate | Total | Infected (%) | (95% CI) |
| --- | --- | --- | --- |
| No Toilet | 438 | 64 (14.6) | (9.9–21.1) |
| Toilet | 1538 | 270 (17.6) | (13.3–22.8) |
| Land cover classification | | | |
| Closed Shrubland | 787 | 184 (23.3) | (17.6–30.3) |
| Open Shrubland | 696 | 83 (11.9) | (8.8–15.9) |
| Sparse Shrubland | 437 | 52 (11.9) | (8.7–16.0) |
| Thickett | 56 | 15 (26.8) | (14.8–43.4) |
| Slope (quintiles) | | | |
| (Lowest) 1 | 390 | 59 (15.1) | (11.0–20.5) |
| 2 | 386 | 72 (18.7) | (12.9–26.3) |
| 3 | 402 | 70 (17.4) | (11.9–24.8) |
| 4 | 404 | 70 (17.3) | (12.4–23.7) |
| 5 | 394 | 63 (16.0) | (11.6–21.6) |

*Computes the proportion of households having access to piped water in the unique community surrounding each participant in the study (**Figure 3**). The Quintile (Q) ranges (min–max) are: Q1: 0–36; Q2: 37–59; Q3: 60–75; Q4: 76–92; Q5: 93–100.

DOI: https://doi.org/10.7554/eLife.33065.003

correspond to a greater than 50% decrease in the odds of infection for individuals living in that community (**Figure 4b**). We also found that the result of our primary analysis (ie effect of community-level piped water coverage) was robust to the exact specification of the community (kernel) evaluating the piped water coverage around each member of the cohort. For example, when we increased the size of the kernel's search radius from 2 km to 3 km then we found that both the continuous (aOR = 0.97, 95% CI: 0.96–0.98, p<0.001) and categorical effect-size estimates (quintile five versus quintile one aOR = 0.12, 95% CI: 0.06–0.24, p<0.001) remained virtually unchanged. Further, we report similar results for a parallel set of analyses (using a Poisson model) to derive risk ratios of infection (**Figure 4—figure supplement 1**, **Supplementary file 2**).

For a comparative exercise, we replaced the mean community-level piped water exposure (2001–2007) exposure variable with the piped water coverage only in the year of the parasitological survey (2007). As anticipated, we find in this model that the resulting risk-reduction is attenuated (by 44%) but nevertheless still significant (aOR = 0.986, 95% CI: 0.98–0.99, p=0.004), providing further empirical support for the analytical approach taken in the longitudinal measurement of mean piped water coverage in the years leading up to the parasitological survey.

As a secondary objective, we analysed the impact of household access to piped water on risk of urogenital schistosomiasis infection. We found that this exposure was independently associated with a significant reduction in the odds of infection (aOR = 0.54, 95% CI: 0.33–0.89, p=0.017; Model 2 of **Table 3**), which was considerably smaller than its community-level counterpart. Once again this reduction was driven mostly by the reduction in risk in girls (aOR = 0.38, 95% CI: 0.19–0.75, p=0.005; Model 2 of **Table 4**), which was substantially larger than the corresponding impact in boys (aOR = 0.76, 95% CI: 0.41–1.42, p=0.38; Model 2 of **Table 4**, see also **Figure 5**). When formally testing this as an interaction effect in the combined-sex model, the reduction in risk was significantly greater for girls (p=0.014).

Our risk-reduction estimate for access to piped water at the household level (aOR = 0.54) is close to the *Schistosoma haematobium* specific aOR of 0.57 generated as part of Grimes et al. systematic review (**Grimes et al., 2014**) comprising 17 previous studies (**Abou-Zeid et al., 2012**; **Al-Waleedi et al., 2013**; **Awoke et al., 2013**; **Dame et al., 2006**; **Dawet, 2012**; **Farooq et al., 1966**; **Howarth et al., 1988**; **Knopp et al., 2013**; **Nworie et al., 2012**; **Reuben et al., 2013**; **Sady et al., 2013**) (**Figure 6**). The revised overall risk-reduction estimate changes marginally after inclusion of our results and rerunning the meta-analysis, to 0.56 (95% CI: 0.45–0.70). The fact that our effect-size estimate for this secondary analysis is close to that obtained from the systematic review and meta-

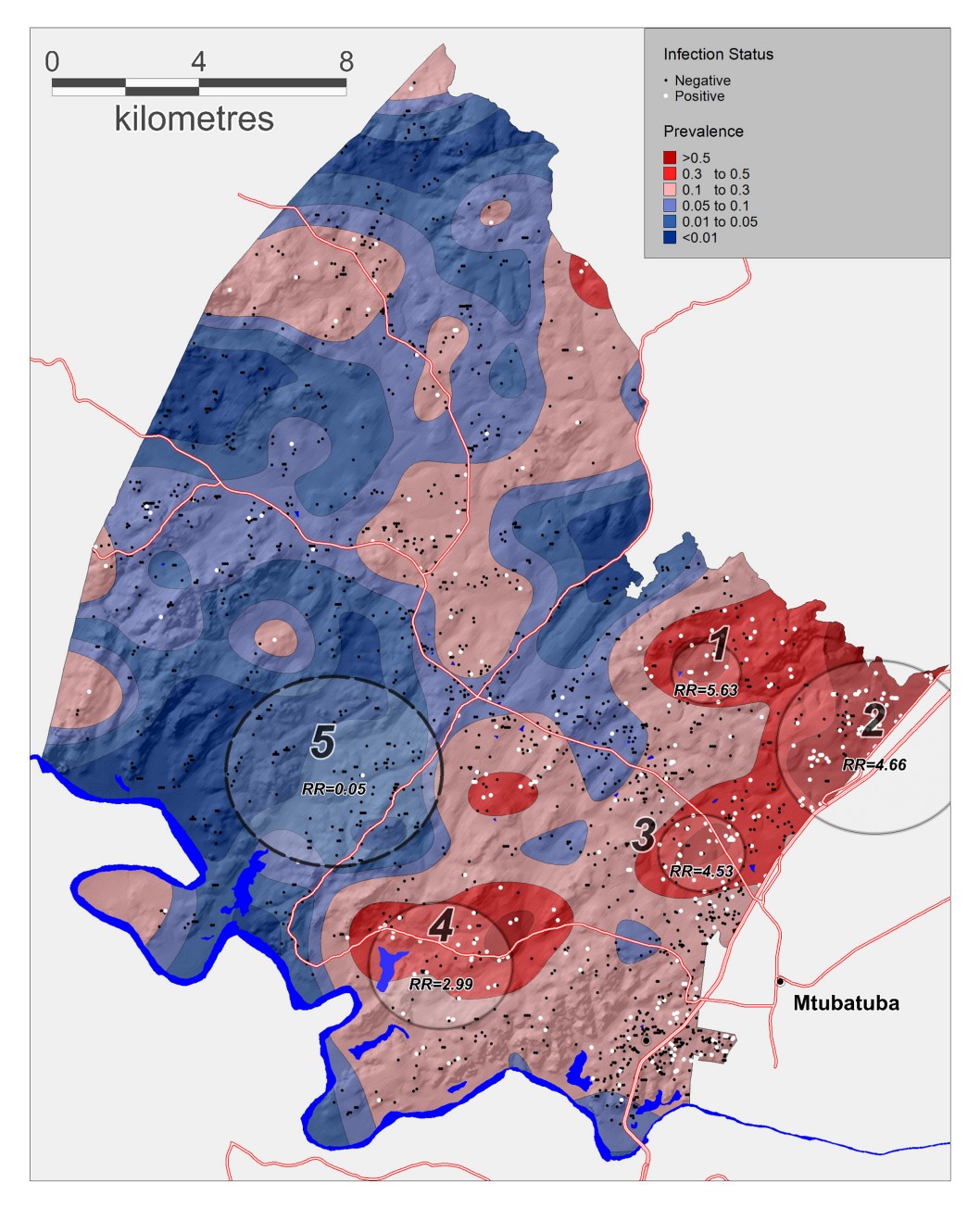

**Figure 2.** Geographical variations in *Schistosoma haematobium* prevalence across the surveillance area obtained by a Gaussian kernel applied to participants' precise household locations. Approximate locations of participants' households are shown (incorporating an intentional random error) with white dots representing an infected child. Superimposed on the map are the clusters of infection independently identified by the Kulldorff spatial scan statistic (cluster 5, low relative-risk; cluster 1–4, high relative-risk). The National Road can be seen running along the Eastern boundary of the surveillance area towards Mozambique.

DOI: https://doi.org/10.7554/eLife.33065.004

analysis (based on 17 data-points), lends further credence to our findings - including the risk-reduction result reported from our primary analysis (described above).

Finally, our IV analysis to address any potential residual confounding confirms all of our main results. Our instrumental variable met the relevance condition of instrumental variable estimation, as shown by the size of the F-test for instrument strength (F-statistic = 55). In the IV analysis using a

linear model, a 1% increase in the coverage of piped water in the surrounding community was independently associated with a 0.48% decrease in the risk of urogenital schistosomiasis infection (Coef= –0.0048, p<0.001; Model 2 of *Supplementary file 2*). This effect-size estimate is larger than the risk reduction observed without the IV, which was 0.30% (Coef = –0.003, p<0.001; Model 1 of *Supplementary file 2*). Household access to piped water was associated with a 15.6% reduction in the risk of infection (Coef = –0.156, p=0.035; Model 4 of *Supplementary file 2*). This effect-size is higher than the estimate in the analysis without the IV, which was 9.2% (Coef = –0.092; p=0.013; Model 3 of *Supplementary file 2*). These findings demonstrate that the reductions in urogenital schistosomiasis risk are not overestimated as a result of other unmeasured variables or residual confounding.

## Discussion

We have linked longitudinal population-based exposures to schistosomiasis outcomes in a large cohort of children to provide a rigorous quantification of the relationship between piped water coverage and the risk of urogenital schistosomiasis infection in a typical rural sub-Saharan African population. This approach provides a strong indicator of the overall impact of the introduction of piped water because it captures both the direct and indirect protection conferred by the scale-up. Our results reveal a large impact of piped water scale-up on urogenital schistosomiasis infection. This impact is substantially greater than previous results had suggested (*Grimes et al., 2014*), based on measurement of access to safe water supplies at the household or individual level. In our study, children living in communities with a high coverage of piped water were eight times less likely to be infected relative to those living in areas with little or no access to piped water. Every 1% increase in the coverage of piped water in the surrounding local community was associated with a corresponding 2.5% decrease in the odds of infection. As a secondary objective, we also provide a robust household-level piped water risk-reduction estimate (aOR = 0.54) after controlling for key socio-economic and environmental predictors of infection. This result is also important, because the relationship between piped water and schistosomiasis infection at the household or individual level would be confounded in different ways by multiple socio-economic and environmental factors, as the recent systematic review of the existing literature points out (*Grimes et al., 2014*). Importantly, we validate the veracity of all of the above findings using an instrumental variable approach to control for the influence of residual unobserved confounding that could have biased our effect-size estimates.

Our results show that nearly 1 in 5 primary school children (median age = 11 years) were infected with urogenital schistosomiasis and nearly 1 in 10 were heavily infected. These levels of urogenital schistosomiasis infection are markedly lower than the overall prevalence of 60–70% recorded in the general area thirty years earlier (*Gear et al., 1980*) and 69% prevalence recorded in a more recent survey undertaken in a neighbouring district at similar altitude (<300 metres above sea level) but with limited access to piped water (*Saathoff et al., 2004*). Importantly, our geographical analyses revealed remarkable spatial heterogeneity in urogenital schistosomiasis prevalence across the relatively small surveillance area. For example, the prevalence of infection was >80% in some communities where the roll-out of piped water had not taken place. At a micro-geographical level, we identified clear spatial clustering of infections around four specific open water collection sites. This heterogeneity in schistosomiasis transmission is consistent with the results of a 2004 study conducted 200 km north of the surveillance area (*Saathoff et al., 2004*).

Another important finding to emerge from the work is that the greater burden of urogenital schistosomiasis infection is borne by boys who had more than twice the prevalence of infection in comparison to girls. In addition, there were clear sex differentials in the pathways of infection and the impact of piped water at a household and community level. Whilst environmental variables (distance to water-body, altitude and landcover class) were highly predictive of infection in both sexes, higher socio-economic status was independently associated with a decrease in the odds of infection for girls but not for boys. At the community level, piped water coverage was associated with significantly greater reductions in the infection risk for girls. Similarly, household access to piped water was associated with a significant reduction in risk in girls but not in boys. These sex differences at both the community and household level could reflect the cultural context in which girls are responsible for

**Table 2.** Significant spatial clusters (either unexpectedly high or low numbers) of *Schistosoma haematobium* infections identified by the Kulldorff spatial scan statistic (p<0.05) across the study area (see *Figure 2*)

| Cluster Number | Radius (km) | Log-Likelihood | P-value | Prevalence (%) | Relative Risk |
|---|---|---|---|---|---|
| 1 | 0.95 | 16.54 | <0.001 | 84.6 | 5.63 |
| 2 | 2.68 | 65.5 | <0.001 | 50 | 4.66 |
| 3 | 1.19 | 30.15 | <0.001 | 49.2 | 4.53 |
| 4 | 1.96 | 14.89 | 0.002 | 41.9 | 2.99 |
| 5 | 2.93 | 17.2 | <0.001 | 0.76 | 0.05 |

DOI: https://doi.org/10.7554/eLife.33065.005

the majority of the domestic water collection and/or a situation where boys swim more often in infected water bodies (*Ngorima et al., 2008*; *Thomassen Morgas et al., 2010*).

The differential in the community/household risk-reduction result for boys suggests that the scale-up of piped water has conferred substantial *indirect* protection to boys through the lowering of the overall parasite reservoir in a particular community. Had we only analysed access to piped water as a single household-level exposure, we would have failed to demonstrate that the scale-up this intervention offered any significant protection to boys. Boys are exposed in the community to free swimming *cercariae* released by snails infected not just by themselves but also by their family members and neighbours. Therefore, in order to fully capture the impact of piped water in reducing schistosomiasis transmission risk, it is crucial to look beyond the household and consider the piped water coverage level in the surrounding local community. This view is also expressed by other authors (*Esrey and Habicht, 1986*). The greater than four-fold ratio in the community-level (aOR = 0.12) versus household-level (aOR = 0.54) risk-reduction estimate provides empirical support for this viewpoint. Whilst no such community-level analyses have previously been conducted for *Schistosoma haematobium*, a study in China found *Schistosoma japonicum* infection rates to be significantly lower in communities where more than 50% of people used 'hygienic lavatories' (*Yang et al., 2009*).

Our study design has several strengths that overcome many of the limitations typically associated with school-based surveys in which the exposures and disease outcomes are measured at the same point in time. By nesting the parasitological survey within a population cohort followed over seven years preceding the measurement of the disease, we utilise longitudinal data collected both on the participants of the parasitological survey, as well as the entire population to provide a rigorous quantification of the impact of the introduction of piped water on risk of urogenital schistosomiasis infection. A key strength of the work is that rather than treating access to piped water as a household/individual level exposure as in previous work (*Abou-Zeid et al., 2012*; *Al-Waleedi et al., 2013*; *Awoke et al., 2013*; *Dame et al., 2006*; *Dawet, 2012*; *Farooq et al., 1966*; *Howarth et al., 1988*; *Knopp et al., 2013*; *Nworie et al., 2012*; *Reuben et al., 2013*; *Sady et al., 2013*), we were able to quantify the clear dose-response in the relationship between community coverage of piped water and risk of infection. In addition, we used a novel spatial methodology to perform a micro-geographical clustering analysis, based on the exact location of the participants' households, to pinpoint clear transmission foci around four open water collection points. Crucially, through the use of high resolution satellite imagery, ancillary environmental data and variables collated as part of the ongoing population cohort, we controlled for the key individual, social, environmental and community determinants of schistosomiasis infection which are vital to quantifying the independent impact of piped water. Notwithstanding the above, we wanted to completely exclude the possibility of residual confounding (on the basis of unmeasured social and environmental factors) leading to an overestimate of the impact of the introduction of piped water on risk of infection. To address this possibility, we performed an instrumental variables (IV) analysis using the year that piped water was first introduced into the community as the instrument. Our results remained robust in this analysis: all effect-size estimates remained significant and increased to some extent in comparison to the main results. These findings demonstrate that the reductions in risk reported in this paper are robust and may be regarded as being marginally conservative.

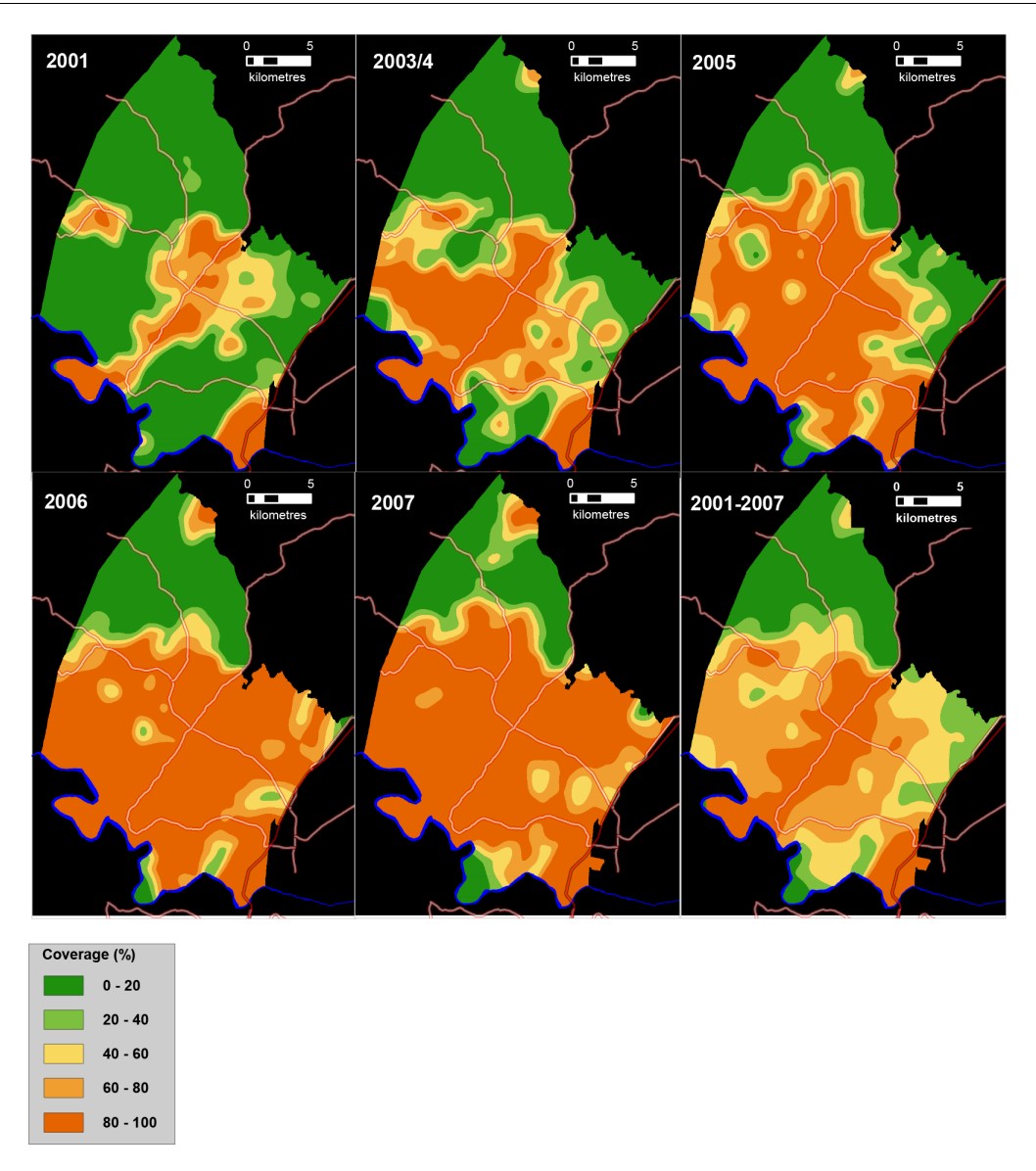

**Figure 3.** Time series of maps showing the coverage of piped water (%) between 2001 and 2007 in the study area (as measured using a Gaussian kernel approach) as well as mean piped water coverage over the full study period (**bottom right**). Main roads are superimposed.
DOI: https://doi.org/10.7554/eLife.33065.006

However, the study has some limitations. The results are based on a single urine sample and as a result we would likely have missed some light infections (*Kosinski et al., 2011*). The true prevalence of infection is therefore probably higher than 16.8% reported in this study. A further limitation is that, while South Africa is a country with near-universal primary school attendance (*Lund, 2011*), we could be missing some children who did not attend school on the day of testing because they have schistosomiasis-related morbidity. This would lead to an under-estimate of the prevalence of urogenital schistosomiasis. In addition, the high mobility of this population (*Dobra et al., 2017*) means that children would be moving between communities, resulting in an ascertainment error leading ultimately to a reduction in the true effect-size, which would bias the result towards the null hypothesis of no piped water impact. In addition, infrequent, ad-hoc schistosomiasis 'test and treat' activities conducted in schools in the highest-risk areas (where piped water coverage was lowest), could

**Table 3.** Logistic regression analysis of the risk factors of *Schistosoma haematobium* infection.
Model 0 gives the univariate results and Model 1 includes all variables in the model. In Model 2, piped water coverage in the immediate community surrounding each participant has been substituted with household-level piped water covariate.

| Covariate | Model 0: Univariate | | | Model 1: Community coverage | | | Model 2: Household access | | |
|---|---|---|---|---|---|---|---|---|---|
| | aOR | (95% CI) | P-value | aOR | (95% CI) | P-value | aOR | (95% CI) | P-value |
| Community piped water quintiles (vs Lowest)† | | | | | | | | | |
| 2 | 0.77 | (0.34, 1.75) | 0.529 | 0.39‡ | (0.23, 0.66) | <0.001 | | | |
| 3 | 0.57 | (0.22, 1.50) | 0.250 | 0.30 | (0.15, 0.59) | <0.001 | | | |
| 4 | 0.27 | (0.10, 0.71) | 0.009 | 0.16 | (0.08, 0.33) | <0.001 | | | |
| 5 | 0.41 | (0.17, 0.99) | 0.048 | 0.12 | (0.06, 0.26) | <0.001 | | | |
| Household access to water (vs No) | | | | | | | | | |
| Yes | 0.96 | (0.56, 1.64) | 0.870 | | | | 0.54 | (0.33, 0.89) | 0.017 |
| Gender (vs Female) | | | | | | | | | |
| Male | 2.24 | (1.64, 3.08) | <0.001 | 2.62 | (1.92, 3.59) | <0.001 | 2.41 | (1.77, 3.28) | <0.001 |
| Age testing | | | | | | | | | |
| Per unit | 1.19 | (1.07, 1.31) | 0.001 | 1.21 | (1.08, 1.36) | 0.001 | 1.18 | (1.06, 1.31) | 0.002 |
| Grade (vs Grade 5) | | | | | | | | | |
| Grade 6 | 0.86 | (0.45, 1.66) | 0.648 | 0.76 | (0.51, 1.11) | 0.149 | 0.77 | (0.47, 1.28) | 0.310 |
| Praziquantel in last 12 months (vs No) | | | | | | | | | |
| Yes | 2.18 | (1.05, 4.52) | 0.038 | 1.27 | (0.60, 2.71) | 0.529 | 1.48 | (0.69, 3.16) | 0.307 |
| Altitude Class (vs < 50) | | | | | | | | | |
| 50–100 | 0.58 | (0.27, 1.22) | 0.147 | 0.47 | (0.23, 0.96) | 0.039 | 0.50 | (0.23, 1.09) | 0.081 |
| 100–150 | 0.23 | (0.09, 0.60) | 0.003 | 0.20 | (0.09, 0.43) | <0.001 | 0.20 | (0.08, 0.51) | 0.001 |
| 150–200 | 0.13 | (0.04, 0.39) | <0.001 | 0.09 | (0.03, 0.25) | <0.001 | 0.11 | (0.04, 0.33) | <0.001 |
| >200 | 0.14 | (0.04, 0.51) | 0.004 | 0.08 | (0.03, 0.29) | <0.001 | 0.12 | (0.03, 0.44) | 0.002 |
| Landcover class (vs Sparse Shrubland) | | | | | | | | | |
| Closed Shrubland | 2.26 | (1.55, 3.28) | <0.001 | 1.56 | (1.05, 2.31) | 0.030 | 2.41 | (1.63, 3.58) | <0.001 |
| Open Shrubland | 1.00 | (0.70, 1.44) | 0.989 | 1.03 | (0.72, 1.47) | 0.863 | 1.44 | (0.96, 2.17) | 0.079 |
| Thickett | 2.71 | (1.28, 5.75) | 0.010 | 1.75 | (0.82, 3.73) | 0.145 | 2.52 | (1.23, 5.18) | 0.012 |
| Slope (square root) | | | | | | | | | |
| per unit | 0.98 | (0.83, 1.16) | 0.818 | 1.02 | (0.89, 1.16) | 0.794 | 0.91 | (0.79, 1.04) | 0.159 |
| Distance to water body (vs < 1 km) | | | | | | | | | |
| 1–2 km | 1.08 | (0.74, 1.58) | 0.667 | 0.78 | (0.56, 1.08) | 0.131 | 0.99 | (0.69, 1.41) | 0.946 |
| 2–3 km | 0.97 | (0.54, 1.75) | 0.928 | 0.72 | (0.47, 1.12) | 0.143 | 1.04 | (0.62, 1.77) | 0.874 |
| >3 km | 0.41 | (0.23, 0.74) | 0.003 | 0.25 | (0.12, 0.49) | <0.001 | 0.44 | (0.24, 0.79) | 0.007 |
| Toilet in household (vs No) | | | | | | | | | |
| Yes | 1.24 | (0.90, 1.72) | 0.186 | 1.24 | (0.87, 1.76) | 0.229 | 1.20 | (0.84, 1.72) | 0.319 |
| Household assets quintile (vs Poorest) | | | | | | | | | |
| 2 | 0.74 | (0.50, 1.09) | 0.123 | 0.88 | (0.60, 1.27) | 0.480 | 0.87 | (0.61, 1.25) | 0.459 |
| 3 | 0.73 | (0.49, 1.09) | 0.127 | 0.78 | (0.51, 1.18) | 0.235 | 0.75 | (0.48, 1.16) | 0.186 |
| 4 | 0.69 | (0.42, 1.14) | 0.143 | 0.81 | (0.47, 1.40) | 0.450 | 0.67 | (0.40, 1.11) | 0.118 |
| 5 | 0.72 | (0.41, 1.24) | 0.228 | 0.80 | (0.48, 1.34) | 0.389 | 0.62 | (0.36, 1.05) | 0.075 |
| Missing | 0.69 | (0.39, 1.21) | 0.194 | 0.84 | (0.40, 1.80) | 0.658 | 0.64 | (0.29, 1.40) | 0.258 |

† Computes the proportion of households having access to piped-water in the unique community surrounding each participant in the study (**Figure 3**). The Quintile (Q) ranges (min–max) are: Q1: 0–36; Q2: 37–59; Q3: 60–75; Q4: 76–92; Q5: 93–100, ‡ Corresponding values for a model in which community-level piped-water coverage is used as a continuous variable: a 1% increase in the coverage of piped-water in the surrounding community, was independently associated with a 2.5% decrease in the odds of a *Schistosoma haematobium* infection (aHR=0.975; 95% CI: 0.966, 0.985; p-value<0.001).

DOI: https://doi.org/10.7554/eLife.33065.009

theoretically bias the effect-size towards the null hypothesis of no piped water impact. However, introduction of such a bias by these activities is unlikely given the large piped water effect-sizes that we report and the fact that only 2% of participants in our survey reported receiving praziquantel in the previous 12 months (which we use as a control variable in the statistical model). In any event, any related 'residual confounding' effect if it did exist, would be controlled for in the IV approach which confirmed all of our main findings.

Schistosomiasis and soil-transmitted helminthiasis affect more than 1.5 billion of the world's poorest people (*Lai et al., 2015*). Without a change in strategy to control these diseases, the population of sub-Saharan Africa will lose an estimated US$3.5 billion of economic productivity every year, which is comparable to recent acute epidemics, including the 2014 Ebola and 2015 Zika epidemics (*Lo et al., 2017*). Our analytical approach (quantifying the combined direct and indirect impact of the scale-up in piped water) has captured a large measurable association between the introduction of piped water and the reduction in risk of urogenital schistosomiasis in a rural African community. This reduction is substantially greater than had been suggested by previous work (*Grimes et al., 2014*), which focused on household or individual level access to safe water. The continued expansion of safe water supplies, decreasing prevalence and highly focal nature of disease transmission, suggest that the disease is retreating and could be vulnerable to eradication in similar African contexts. Recent modelling work indicates that the current WHO treatment goals (which aim to mass-treat 75% of at-risk children) should be successful in eradicating schistosomiasis transmission if continued for a decade in areas of low to moderate transmission (*Anderson et al., 2015*). However, once mass treatment is suspended, the prevalence of infection can rebound to pre-control levels over a period of 25–30 years (*Gurarie et al., 2015*), emphasising the need for a fully integrated approach that includes the provision of safe water supplies, community health education as well as the possibility of focal snail control (*Knopp et al., 2013*; *King et al., 2015*).

The overall moderate levels of infection (significantly lower than historical surveys (*Gear et al., 1980*) combined with the extreme inter-host and geographical heterogeneity as well as ever increasing access to safe water suggest that sustained mass treatment campaigns in such populations could drive schistosomiasis towards low levels of endemicity (*Crompton and WHO, 2006*). Our results emphasise the need to rapidly increase the provision of safe water supplies to schistosomiasis endemic areas, where the resulting impact will extend well beyond prevention of transmission of schistosomiasis to diarrhoeal diseases, for example (*Esrey and Habicht, 1986*); as well as having substantial related economic benefits (*Lo et al., 2017*). Scale-up in the provision of safe water and sanitation, along with the small cost of drugs combined with the renewed commitment to increase funding for neglected tropical diseases, including schistosomiasis, means that there is a real opportunity to push back the disease from areas where it has long been endemic.

## Materials and methods

### Schistosomiasis in South Africa

In South Africa, *Schistosoma haematobium* is endemic over an area of 320 000 $km^2$ including parts of six of the country's nine provinces to the east of latitude 26°E (*Doumenge et al., 1987*). Highest prevalence and intensities occur in the lower lying areas of Limpopo, Mpumalanga and KwaZulu-Natal provinces. Estimates of the number of children infected have been based largely on the 38 year old Atlas of Bilharzia (*Gear et al., 1980*), and range from 1.5 million to 4 million (*Moodley et al., 2003*). The most frequently infected age group is 5 to 15 years after which prevalence declines to a generally low level in adulthood. Morbidity of the urinary tract, which is greatest in children with high worm and egg burdens, follows a similar pattern. Considerable gynaecological morbidity, also due to *Schistosoma haematobium* and known as female genital schistosomiasis, has been demonstrated in KwaZulu-Natal (*Hegertun et al., 2013*) and is probably present over much of the endemic area. *Schistosoma haematobium* has a significantly wider distribution in South Africa than its congener *Schistosoma mansoni* which causes intestinal or rectal schistosomiasis (*Appleton and Madsen, 2012*). KwaZulu-Natal is the best researched of the endemic provinces in

**Table 4.** Multivariable model examining the socio-demographic predictors of *Schistosoma haematobium* infection stratified by gender.

Model 1 includes all variables in the model. In Model 2, piped water coverage in the immediate community surrounding each participant has been substituted with household-level piped water covariate.

| Covariate | Model 1: Community-level coverage of piped water | | | | | | Model 2: Household level access to piped water | | | | | |
| | Females | | Males | | | | Females | | Males | | | |
| | aOR§ (95% CI) | P-value | aOR§ (95% CI) | P-value | | | aOR§ (95% CI) | P-value | aOR§ (95% CI) | P-value | | |
|---|---|---|---|---|---|---|---|---|---|---|---|---|
| Community piped water quintiles (vs Lowest)† | | | | | | | | | | | | |
| 2 | 0.24 (0.10–0.57)‡ | 0.002 | 0.56 (0.35–0.90)‡ | 0.017 | | | | | | | | |
| 3 | 0.21 (0.08–0.55) | 0.002 | 0.37 (0.17–0.78) | 0.011 | | | | | | | | |
| 4 | 0.16 (0.06–0.45) | 0.001 | 0.16 (0.08–0.35) | <0.001 | | | | | | | | |
| 5 | 0.07 (0.02–0.20) | <0.001 | 0.17 (0.08–0.36) | <0.001 | | | | | | | | |
| Household access to water (vs No) | | | | | | | | | | | | |
| Yes | | | | | | | 0.38 (0.19–0.75) | 0.005 | 0.76 (0.41–1.42) | 0.379 | | |
| Age at Testing | | | | | | | | | | | | |
| Per unit | 9.62 (1.62–57.22) | 0.014 | 1.24 (1.08–1.42) | 0.003 | | | 6.85 (1.56–30.16) | 0.011 | 1.20 (1.06–1.37) | 0.005 | | |
| Age$^2$ | | | | | | | | | | | | |
| Per unit | 0.92 (0.85–0.99) | 0.024 | | | | | 0.93 (0.88–0.99) | 0.017 | | | | |
| Toilet in household (vs No) | | | | | | | | | | | | |
| Yes | 1.59 (0.88–2.88) | 0.121 | 1.12 (0.70–1.86) | 0.667 | | | 1.26 (0.75–2.14) | 0.371 | 1.11 (0.66–1.86) | 0.687 | | |
| Household Assets quintile (vs Poorest) | | | | | | | | | | | | |
| 2 | 0.46 (0.24–0.90) | 0.026 | 1.27 (0.75–2.08) | 0.344 | | | 0.55 (0.30–0.99) | 0.045 | 1.24 (0.76–2.04) | 0.376 | | |
| 3 | 0.35 (0.13–0.89) | 0.032 | 1.21 (0.72–1.99) | 0.453 | | | 0.35 (0.15–0.84) | 0.019 | 1.19 (0.71–1.98) | 0.502 | | |
| 4 | 0.42 (0.18–1.00) | 0.061 | 1.18 (0.55–2.28) | 0.646 | | | 0.40 (0.18–0.91) | 0.029 | 0.94 (0.46–1.94) | 0.885 | | |
| 5 | 0.66 (0.32–1.37) | 0.307 | 0.90 (0.43–1.75) | 0.769 | | | 0.54 (0.25–1.17) | 0.115 | 0.71 (0.34–1.44) | 0.337 | | |
| Missing | 0.57 (0.20–1.66) | 0.288 | 1.10 (0.38–2.94) | 0.815 | | | 0.40 (0.12–1.35) | 0.135 | 0.91 (0.34–2.42) | 0.845 | | |

§All estimates simultaneously adjusted for landcover class, distance to water, altitude, slope, treatment in the last 12 months and school grade.

† Computes the proportion of households having access to piped-water in the unique community surrounding each participant in the study (**Figure 3**). The Quintile (Q) ranges (min–max) are: Q1: 0–36; Q2: 37–59; Q3: 60–75; Q4: 76–92; Q5: 93–100

DOI: https://doi.org/10.7554/eLife.33065.010

South Africa and historically *Schistosoma haematobium* prevalence and intensities have reached 70% or more and 200 eggs/10 ml (geometric mean) respectively between the coast and an altitude of 300 meters above sea level. Both prevalence and intensity have been observed to decrease with increasing altitude to around 20% and <10 eggs/10 ml above 800 metres above sea level. *Schistosoma mansoni* is less common and limited to patches of transmission, mostly in the province's lowlands (**Appleton and Kvalsvig, 2006**). In the coastal plain of north-eastern KwaZulu-Natal (where the survey was conducted), *Schistosoma mansoni* is largely absent (**Gear et al., 1980**), for reasons discussed in detail elsewhere (**Appleton and Madsen, 2012**).

## Study area

The Africa Health Research Institute (AHRI) runs one of Africa's largest ongoing population-based cohorts in the Umkhanyakude district of the northern KwaZulu-Natal province (**Figure 7**). Approximately 250 km north of the city of Durban, the surveillance area is 438 km$^2$ in size with a population of approximately 90,000 isiZulu-speaking people and 10,000 households. AHRI has collected detailed demographic, socio-economic and behavioural data every four to six months from residents in the surveillance area since 2000 (**Tanser et al., 2008**). All households have been positioned to an accuracy of <2 m using differential global positioning systems (**Tanser et al., 2001**).

The surveillance area is designated as a coastal lowland (<300 metres above sea level) and falls within the endemic schistosomiasis belt of South Africa (**Gear et al., 1980**; **Moodley et al., 2003**).

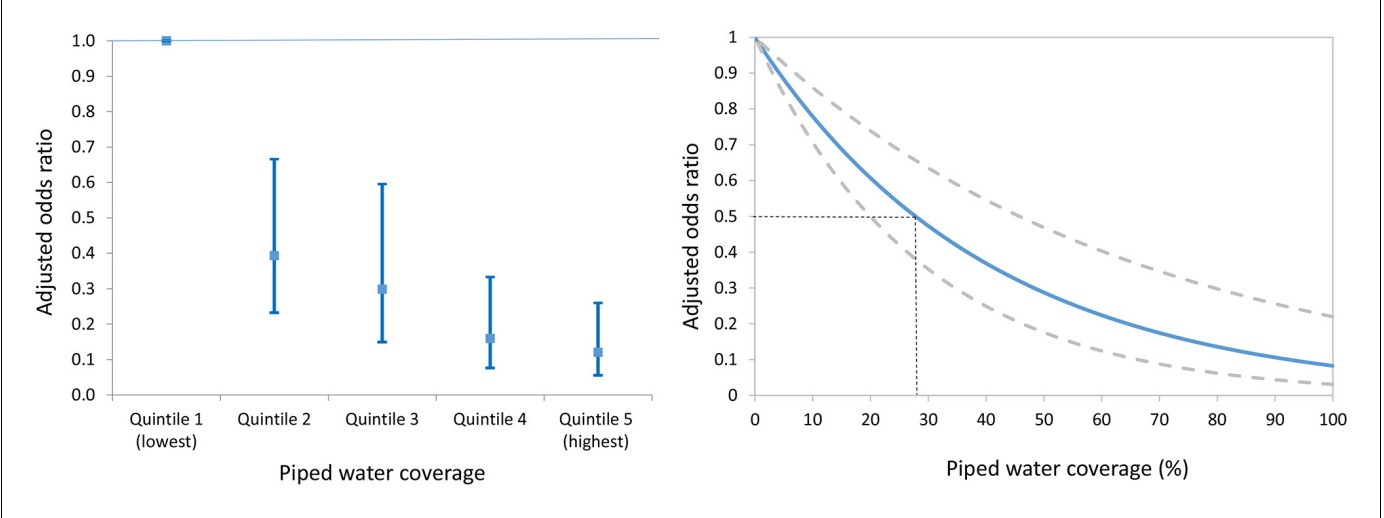

**Figure 4.** Adjusted odds ratio of *Schistosoma haematobium* infection (95% CI) by piped water coverage in the surrounding local community - coverage quintile (**Left**) and continuous piped water coverage (**Right**). The piped water coverage measure (2001–2007) is derived using a Gaussian kernel to calculate the proportion of all households in the unique local community surrounding each participant having access to piped water (*Figure 3*). Odds ratios are adjusted for age, sex, household assets, toilet in household, landcover class, distance to water body, altitude, slope, treatment in the last 12 months and school grade. Standard errors are adjusted for clustering by school and grade.

DOI: https://doi.org/10.7554/eLife.33065.007

The following figure supplement is available for figure 4:

**Figure supplement 1.** Results of a parallel analysis using a Poisson regression.
DOI: https://doi.org/10.7554/eLife.33065.008

The medication praziquantel is available in some of the local clinics to treat schistosomiasis symptoms. In the period leading up to the parasitological survey, there were no mass treatment campaigns, but the local Department of Health did undertake occasional, ad-hoc 'test and treat' activities in a small number of schools in high-risk areas at different points in time. The population has been exposed to a devastating HIV epidemic. In 2010, the HIV prevalence was 24% among all adults and approaching 60% in some age-groups (*Tanser et al., 2013*; *Vandormael et al., 2018*).

## Sample size

The sample size calculation was based on the number of participants required to test the impact of access to piped water on infection with *Schistosoma haematobium* stratified by sex, as well as provide good geographical coverage of the surveillance area. Given two grades sampled at each of the 33 schools (50 pupils per grade) in the study area and an assumption of 70% consent and 90% linkage to the population cohort, we would attain an estimated sample size of 2079. We then assumed that the piped water exposure occurred in 50% of participants and that 30% of pupils tested for *Schistosoma haematobium* were infected. This prevalence estimate was based on historical parasitological surveys in surrounding areas (*Gear et al., 1980*), more recent surveys in the province of KwaZulu-Natal (*Jinabhai et al., 2001*; *Taylor et al., 2001*) and expected reductions in prevalence as a result of the piped water roll-out in the study area during the preceding decade. Under these assumptions, we would attain 99% power to detect a relative risk of ≥1.5 (95% CI) in boys and girls respectively as well as provide good geographical coverage of the study area to facilitate identification of micro-geographical clustering of infections.

## Ethics approval

Ethical clearance for the study was obtained from the Biomedical Research Ethics Committee of the University of KwaZulu-Natal (reference number E165/05). Prior to testing, we were required to obtain both written informed consent from parents and written assent from children participating in the survey (described further below).

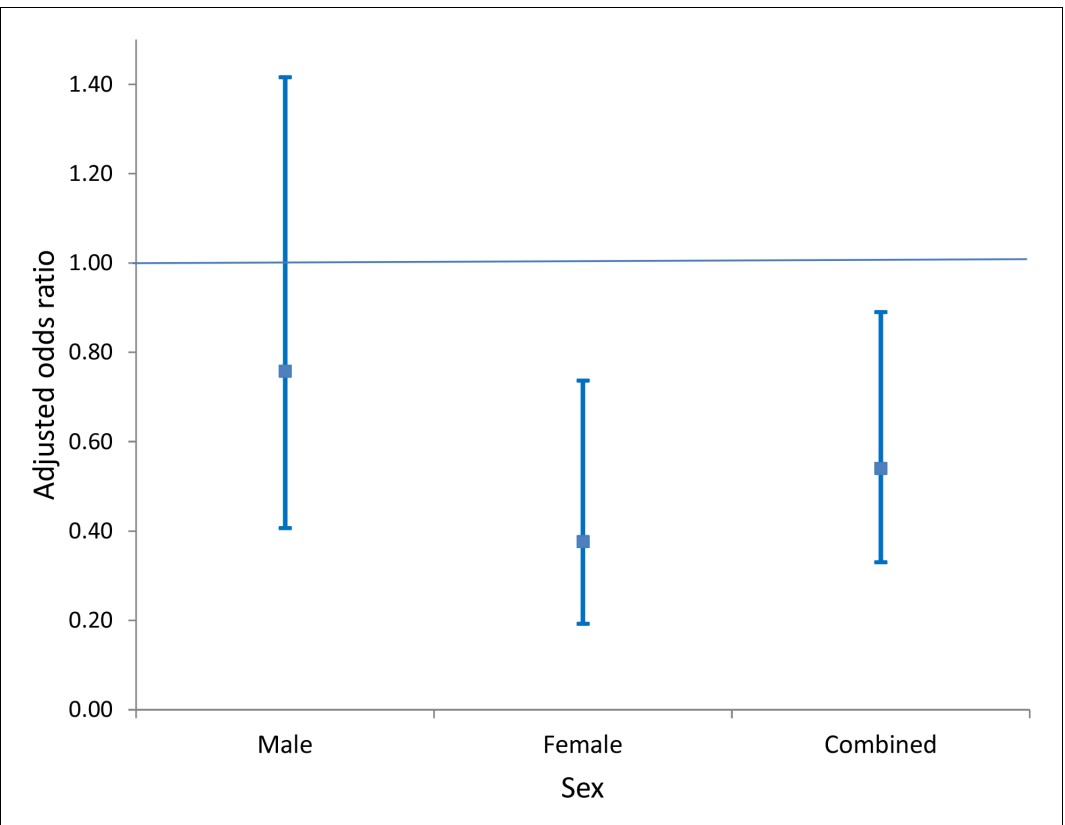

**Figure 5.** Comparison of adjusted odds of *Schistosoma haematobium* infection (95% CI) in participants living in households with access to piped water (relative to participants without household-access to piped water) by sex. The resulting risk estimates are adjusted for age, household assets, toilet in household, landcover class, distance to water body, altitude, slope, treatment in the last 12 months and school grade. Standard errors are adjusted for clustering by school and grade.

DOI: https://doi.org/10.7554/eLife.33065.011

## The *Schistosoma haematobium* survey

We conducted a parasitological survey among 2105 consenting children (median age = 11) in their fifth and sixth year of primary school at all of the 33 primary schools in the surveillance area between March and December 2007. We chose to sample these two school grades based on likely highest levels of *Schistosoma haematobium* infection as well as to facilitate a follow-up survey one year later in the same cohort of children (before completion of their primary school education at the end of year 7). Prior to the study, we initiated an extensive community preparation process that included consultations with headmasters and the governing bodies of participating schools, and delivered presentations to parents and teachers at school meetings. We distributed consent forms to the parents through their children the day before testing. Assent forms were signed by the children on the day of testing. We excluded children who did not have both parental informed consent and child assent from the parasitological survey.

After receiving the written consent and assent forms, we gave participants a cup of juice to drink and a 500 ml honey jar to provide a full urine sample between the hours of 10h00 and 12h00. We labelled each urine sample with a unique bar code and performed a rapid specimen test for the presence of micro haematuria using urinalysis reagent strips (Bayer Uristix). Following national guide-lines, we treated those with detectable micro haematuria with a single oral dose of praziquantel (Biltricide, Bayer) on the basis of body weight (40 mg per kg) (*Saathoff et al., 2004*; *WHO Expert Committee, 2002*). We sent the urine samples for microscopic analysis (described in detail below) and participants found to be falsely negative at baseline were treated at a follow-up visit. Treatment was administered and strictly supervised by qualified nurses. After at least 30 days, we revisited the

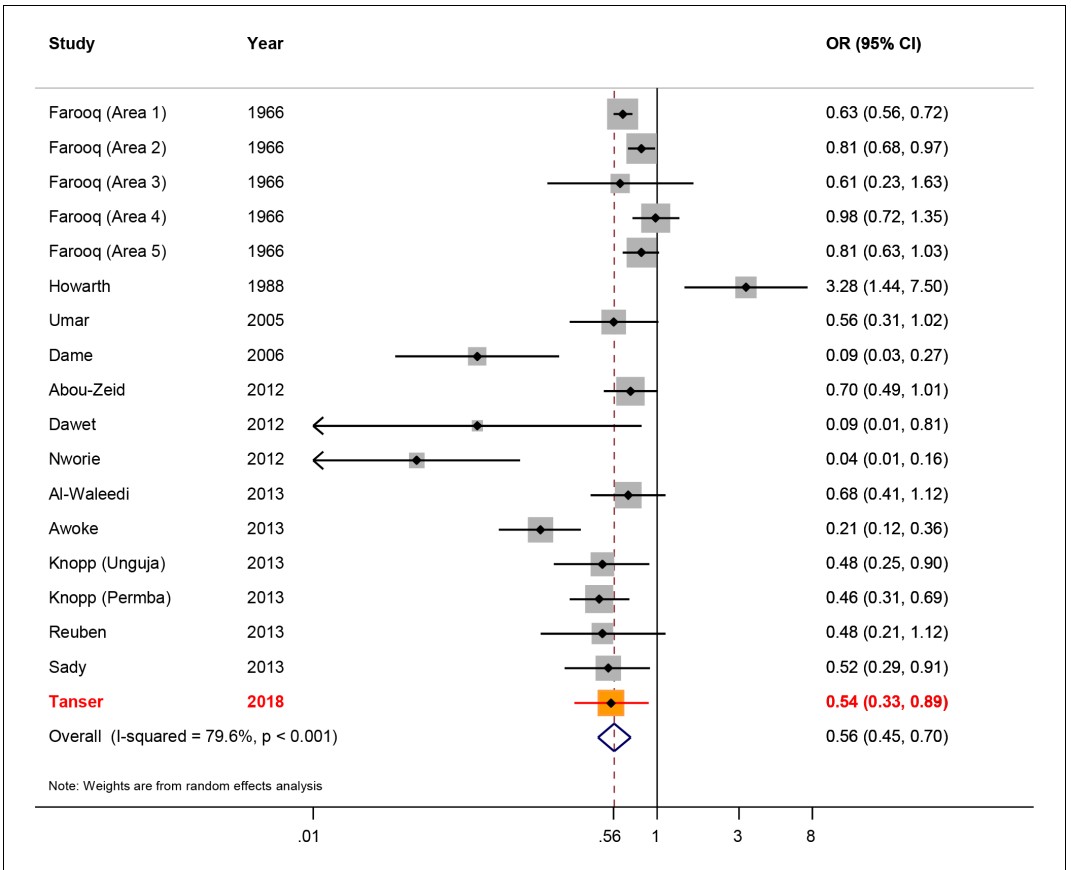

**Figure 6.** Forest plot of *Schistosoma haematobium* infection according to household/individual level access to a safe water. The data are taken from Grimes et al. systematic review (***Grimes et al., 2014***), based on 17 data-points (***Abou-Zeid et al., 2012***; ***Al-Waleedi et al., 2013***; ***Awoke et al., 2013***; ***Dame et al., 2006***; ***Dawet, 2012***; ***Farooq et al., 1966***; ***Howarth et al., 1988***; ***Knopp et al., 2013***; ***Nworie et al., 2012***; ***Reuben et al., 2013***; ***Sady et al., 2013***) to which we have added our study results. The sizes of the squares represent the weight given to each study, the rhombus is the effect-size with the black lines representing the 95% confidence intervals. The overall rhombus represents the combined effect-size, with the results of this study shown in red.
DOI: https://doi.org/10.7554/eLife.33065.012

schools and gave participants (who were treated at the first visit) a second dose of praziquantel if micro haematuria was again detected. We administered a brief questionnaire after testing and asked participants if they had been treated with praziquantel in the last 12 months.

## Laboratory procedures

We aliquoted the urine sample into two 10 ml sub-samples and diluted with a preservation solution (Formalin Methiolate concentration), and uniquely bar-coded each aliquot. The microscopic analysis was done using the filtration technique as described by the World Health Organization (***WHO, 1991***). All eggs on the slides were counted and expressed as eggs per centilitre (10 ml) of urine. Specimens of less than 10 ml were measured before filtration and the number of eggs per 10 ml calculated. Samples with <50 eggs/10 ml of urine specimen were categorised as light infection while those with ≥50 eggs/10 ml of urine specimen were categorised as heavy infection (***WHO Expert Committee, 2002***).

## Derivation of community-level piped water coverage

We used all five separate population-based surveys (2001, 2003/2004, 2005, 2006, 2007) conducted up to the year of parasitological survey (2007) in the surveillance area to measure access to piped water (***Tanser et al., 2008***). There were on average 10,267 households per survey. All

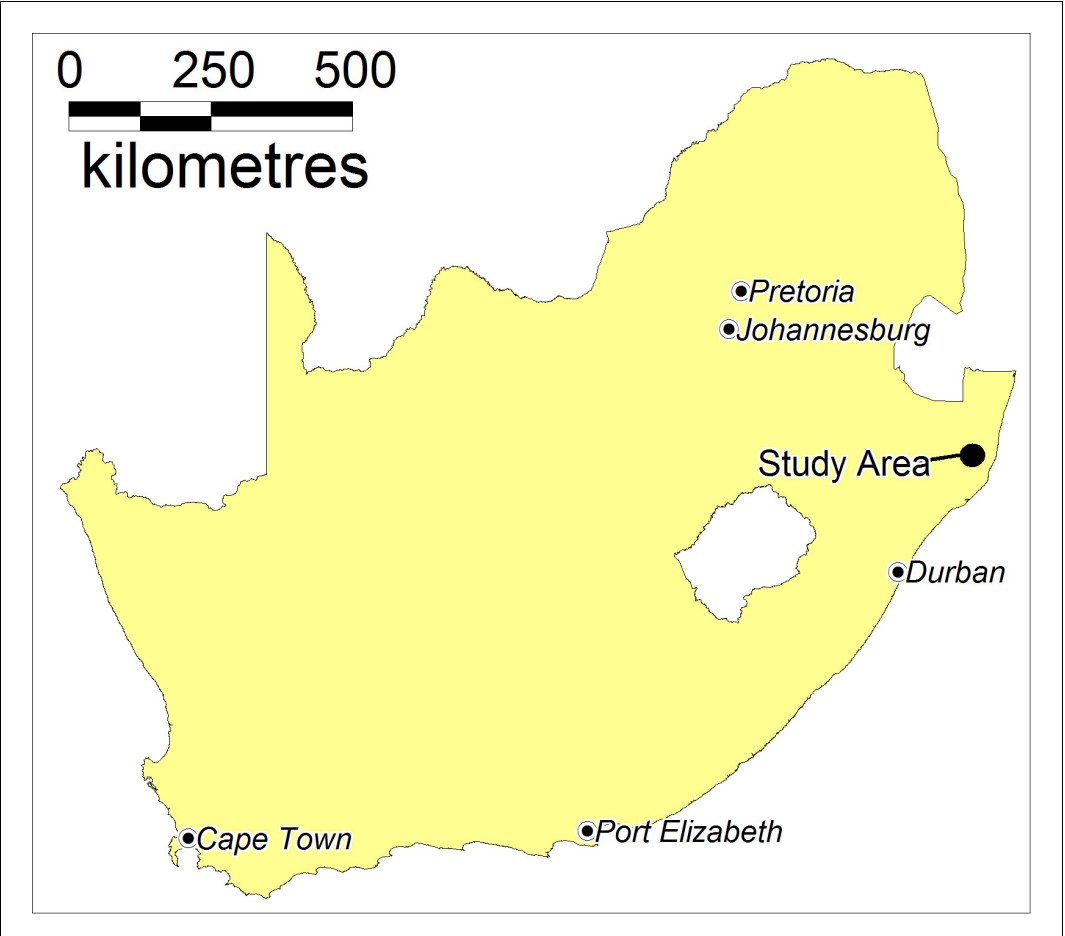

**Figure 7.** Location of the study area in South Africa.
DOI: https://doi.org/10.7554/eLife.33065.013

households in the surveillance area were asked; 'which is the most often used source of drinking-water in this household?' The household key informant was asked to select the 'most common' source from a list of 11 different sources of water listed. Households were deemed to have access to piped water if they reported their most common source of drinking water to be piped water delivered either directly to the household or from a public tap.

We used this population-based data to derive a continuous measure of the mean piped water coverage in the unique community surrounding each participant's household (between 2001 and 2007). This community-level piped water measure was first constructed by means of a moving two-dimensional Gaussian kernel of 2 km search radius (*Tanser, 2006*; *Tanser et al., 2009*). For comparative purposes, we also produced an equivalent analysis using a kernel search radius of 3 km. First, the piped water data were superimposed on a geographic representation of the surveillance area consisting of a grid of 30m × 30 m pixels. Next, the kernel moves systematically across the grid and calculates a Gaussian weighted estimate of the piped water coverage for the unique neighbourhood around each and every pixel on the grid. The methodology therefore provides a sensitive and realistic estimate of piped water coverage in each household's unique surrounding 'virtual community'. We located each participant to an exact homestead of residence and extracted the corresponding mean piped water coverage (2001–2007) exposure estimate in the surrounding local community.

The community-level piped water coverage exposure measure (2001–2007) fully captures the dynamic nature of the roll-out and availability of piped water in the seven years leading up to the parasitological survey. This is important for two reasons. Firstly, it ensures that the exposure measurement (piped water coverage) precedes the measurement of the disease outcome (urogenital

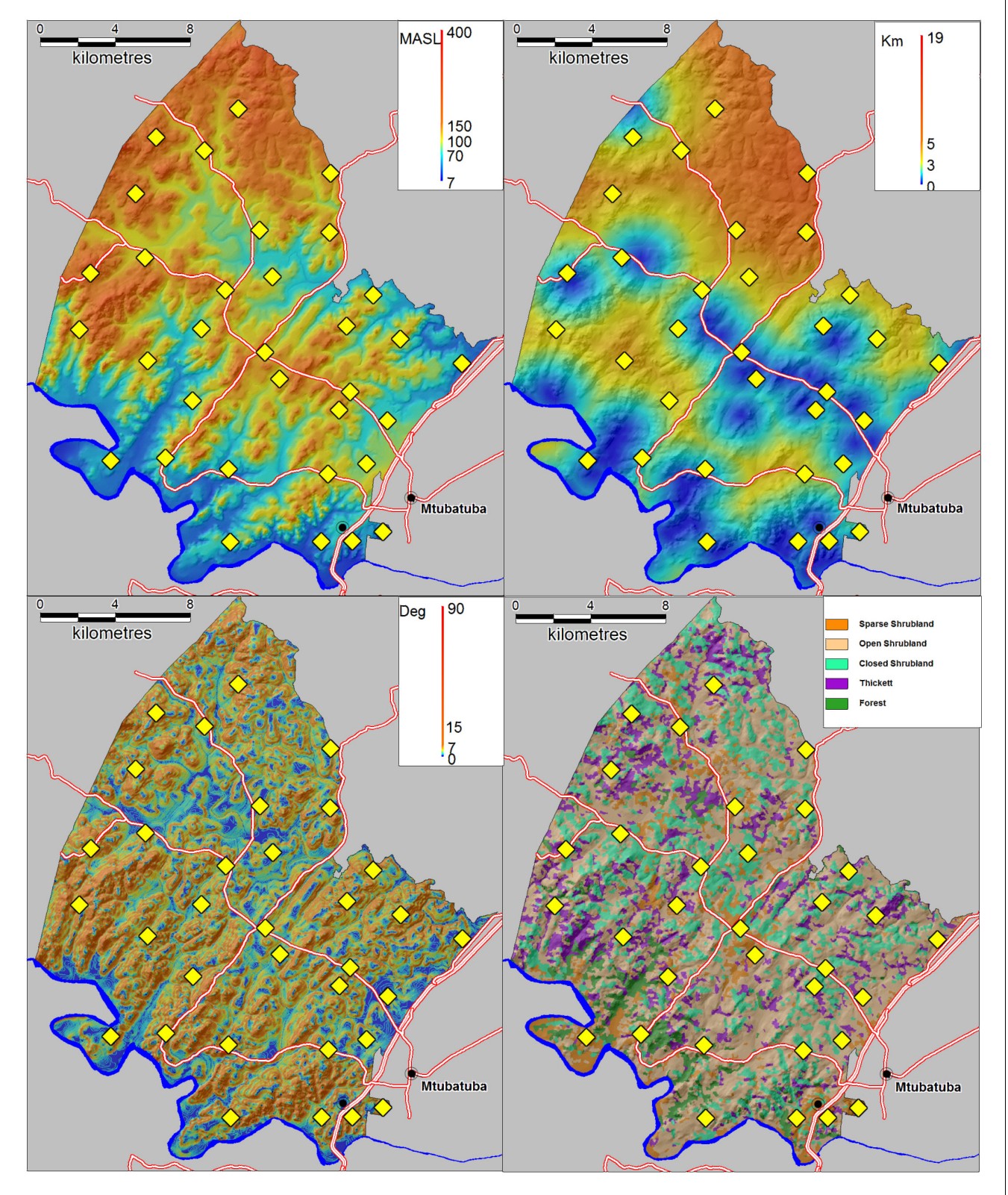

**Figure 8.** Environmental control variables used in the statistical analysis with locations of 33 schools superimposed (yellow diamonds). (**Top left**) Altitude in metres above sea level (MASL) (**Top right**) Distance to nearest water body (km) (**Bottom left**) Slope (degrees) (**Bottom right**) Satellite-derived landcover classification.

DOI: https://doi.org/10.7554/eLife.33065.014

schistosomiasis). This is vital for a chronic infection like schistosomiasis in which children may have been infected for several years before the disease outcome was measured. Secondly, there are occasional instances of water schemes being implemented in the surveillance area but not working consistently for periods of time. In this regard, a community having access to piped water in one year does not necessarily guarantee access to piped water in the following year. Thus, our community-level piped water coverage measure fully captures the vagaries of access to piped water in the years leading up to the parasitological survey.

## Statistical analysis

Of the 2105 children who participated in our parasitological survey, 94% (N = 1,976) could be linked to their corresponding records in the population-based cohort and were included in the statistical analysis. From this we obtained the precise geographical coordinates of each participant's place of residence as well as detailed individual-level and household-level predictors of infection. Children were considered to be infected with *Schistosoma haematobium* on the basis of the microscopy result. To facilitate direct comparison with previous work (*Grimes et al., 2014*), we used a logistic regression as our main statistical model to quantify the impact of access to piped water both at the community and household level respectively, whilst simultaneously controlling for the key environmental and socio-economic predictors of urogenital schistosomiasis infection (described below). However, in light of the expected relatively high prevalence of infection, we also report results for a Poisson regression model to obtain risk ratios of *Schistosoma haematobium* infection. We undertook the logistic and Poisson regression analyses in Stata 14.1, and utilised the *svy* suite of commands to account for the clustering of participants in the 33 schools (two grades) from 66 primary sampling units (*Kreuter and Valliant, 2007*). The failure to correct for clustering violates the independence assumption and can lead to distorted parameter estimates and standard errors (*Guo and Zhao, 2000*). We stratified our analyses by sex to test for any differential impacts in pathways of infection between males and females.

Finally, we formally compared the resulting risk-reduction estimate from our secondary analysis (ie exposure to piped water at the household level) to the effect-size estimate generated as part of Grimes et al. systematic review and meta-analysis (*Grimes et al., 2014*) comprising previous studies (*Abou-Zeid et al., 2012*; *Al-Waleedi et al., 2013*; *Awoke et al., 2013*; *Dame et al., 2006*; *Dawet, 2012*; *Farooq et al., 1966*; *Howarth et al., 1988*; *Knopp et al., 2013*; *Nworie et al., 2012*; *Reuben et al., 2013*; *Sady et al., 2013*). To do this, we captured the results (i.e., sample sizes, odds ratios, and 95% confidence intervals) from the 17 data-points along with the results from our analysis described above. We then ran a random-effects meta-analysis on this data using the *metan* package in Stata (*Harris et al., 2008*).

## Spatial analysis

We used two different and complementary approaches to assess the spatial distribution of urogenital schistosomiasis infection in the study population. Firstly, we used the Gaussian kernel methodology described above (using the precise location of every participant's household) to produce a prevalence map that varies in continuous geographical space. Using this approach, a median of 166 (IQR = 114–248) participants were evaluated for each cell's unique neighbourhood (search radius = 3 km).

Secondly, we used a Kulldorff spatial scan statistic (*Kulldorff et al., 2005*; *Kulldorff, 1997*) to formally identify and map micro-geographical clusters of unusually high numbers of urogenital schistosomiasis infections that were unlikely to have arisen by chance. The Kulldorff spatial scan statistic is implemented within the SaTScan spatial cluster detection programme (*Kulldorff, 2006*). We have used this technique used in our previous work in this population to identify spatial clusters of HIV infections (*Tanser et al., 2009*). A spatial scan statistic is a cluster detection test that is able to both detect the location of clusters and evaluate their statistical significance without problems associated with multiple testing. This is done by gradually scanning a window across space. The statistical theory behind spatial scan statistics is described elsewhere (*Kulldorff et al., 2005*). Briefly, the spatial scan statistic imposes a circular window on a map and it allows the centre of the circle to move systematically across the study region. For any given position of the centre, the size of the circle changes continuously so that it can take any value between zero and a specified maximum value. For

every potential cluster, a likelihood ratio test statistic is used to determine if the number of urogenital schistosomiasis cases within the potential cluster was higher than expected. Expected numbers of cases were calculated on the basis of the null hypothesis of spatial randomness by assuming that the number of cases in each circle is an independent Bernoulli random variable with constant prevalence. The circle with the maximum likelihood is defined as the most likely cluster (implying that it is least likely to have occurred by chance). The maximum observed value of the test statistic for each possible cluster is then compared to the overall distribution of maximum values (*Kulldorff, 2006*). The p-value of the statistic is obtained through Monte Carlo hypothesis testing (9999 iterations), where the null hypothesis of no clustering is rejected if the simulated p-value is <0.05. We allowed the clusters to overlap by <50% and set the maximum search radius of the circle to be 3 km to facilitate comparison with the Gaussian kernel smoothing approach described above.

## Adjustment for observed social and environmental predictors of infection

Schistosomiasis is a complex parasitic disease characterised by key environmental and social predictors which vary in time and space (*Shiff, 2001*). It is therefore not possible to accurately quantify the independent impact of piped water without taking into account the extent to which the local environmental and social conditions in a particular area are conducive to transmission. To control for key environmental differences, we extracted altitude, slope and distance to the nearest water body for every household in the surveillance area (*Figure 8*). To extract altitude and slope, we used the 30 m raster digital elevation model obtained from the South African Surveyor General. Distance to the nearest water body was derived using a combination of 1:50 000 digital data and additional water bodies digitised from 1:10 000 orthorectified aerial photographs. We also used LandSat TM bands 1–7 to map five different ecological units occurring in the surveillance area (*Figure 8*). The derivation of this landcover classification is described in further detail below. For socio-demographic control variables, we used age, sex, praziquantel in the last 12 months, school grade, presence of the toilet in the household and a household socio-economic assets index (generated in a principal component analysis based on household ownership of 24 assets).

To map the ecological units across the study area, we obtained 2007 LandSat multi-spectral, high-resolution (30 m × 30 m) satellite imagery which captures reflectance values at seven regions of electromagnetic spectrum. The data were corrected for radiometric and geometric distortions (*Richards and Jia, 2006*). We used an iterative self-organising unsupervised classifier ISOCLUST implemented within IDRISI Taiga (Clark University, MA, USA) to classify the image into five landcover classes. The clustering procedure implements a histogram peak cluster analysis technique that searches for frequency peaks higher than those of its immediate neighbours. We used the 'broad level of generalisation' option which allocates all pixels to major frequency peaks in the multidimensional histogram (minor frequency peaks are ignored) (*Eastman, 2009*). The module uses a true maximum likelihood clustering procedure that leads to a more efficient and accurate placement of clusters than either random or systematic placement. These landcover classes were subsequently described using a combination of orthorectified 1:10 000 aerial photographs and physical on-the-ground inspection of 40 sites.

A common problem in the use of high resolution satellite imagery is the high degree of spectral variation and spectral scatter within small areas leading to inconsistencies in the classification process. In addition, from a disease exposure perspective, it makes sense to consider an area greater than immediate the 30 m × 30 m area around the place of residence. We therefore employed an image segmentation procedure in IDRISI Taiga that uses a watershed delineation approach to partition the image into a series of ecological units (segments) on the basis of spectral similarity (using a 30 pixel similarity tolerance) (*Eastman, 2009*). Each segment in the resulting image was then classified as one of the five broad landcover classes identified in the previous classification result (using a majority rule). This process improves the accuracy of the initial classification and producing a smoother classification result while preserving the boundaries between segments (*Richards and Jia, 2006*). Two of the resulting landcover classes were then combined into a single class due to their similarity and small numbers of individuals living within them.

## Adjustment for unobserved social and environmental predictors of infection

To test the veracity of our findings, we extended the statistical analysis to address the possibility of residual confounding (via unobserved factors) by carrying out an instrumental variable (IV) analysis. IV analyses allow a quasi-random exposure assignment by isolating the random component of an exposure that is not completely randomly assigned (*Baiocchi et al., 2014*; *Bärnighausen et al., 2017*; *Angrist et al., 1996*). IV analyses require a variable, the instrument, that is significantly associated with the exposure of interest (here: piped water exposure) but is not independently associated with the outcome of interest (here: urogenital schistosomiasis infection). We used the year when piped water was first provided in a traditional Zulu ward, the *isigodi*, as an instrumental variable to isolate the exogenous or as-good-as random variation in the exposure of interest (piped water). A confluence of factors, each of which is fast-changing in this community, led to the delivery of piped water to a particular local area at a particular point in time, including complex political constellations at the local and higher levels, availability of construction capacity, and presence of particular household members. The joint presence of this constantly-changing multitude of factors leading to piped water delivery at a particular time-point was likely as-good-as random regarding other potential determinants of infection, such as wealth or coverage with health services (*UWP Engineers, 2002*). It is thus unlikely to be independently correlated with the outcome (urogenital schistosomiasis infection) and likely meets the exclusion restriction (*Angrist et al., 1996*). For the IV analysis, we used a one-stage maximum likelihood estimation with a linear probability model as implemented in the *ivregress* command in Stata 14.1 (*Baum, 2006*). Specifically, we added to the model our instrumental variable—the year that piped water was introduced into the Zulu ward—along with the same environmental and socio-economic control variables used in our primary analysis. To facilitate direct comparison with the IV analysis, we repeated our main analysis using a linear probability model.

### Role of the funding source

The funders (The International Collaboration in Infectious Disease Research and the Wellcome Trust) had no role in the design and conduct of the study, interpretation of the data, or approval of the report. The corresponding author had access to all study data and made the final decision to submit for publication.

## Acknowledgements

The study was funded through the National Institutes of Health via the International Collaboration in Infectious Disease Research (ICIDR). Funding for the Africa Health Research Institute's population-based cohort was received from the Wellcome Trust, UK. The authors are indebted to the School-Health team at Hlabisa Hospital for their invaluable assistance in conducting the parasitological survey. The authors wish to express their grateful thanks to Colleen Archer (University of KwaZulu-Natal) for conducting the microscopic analysis and Colin Newell for database support.

## Additional information

### Funding

| Funder | Author |
| --- | --- |
| National Institutes of Health | Frank Tanser Christopher Appleton |
| Wellcome Trust | Frank Tanser |

The funders had no role in study design, data collection and interpretation, or the decision to submit the work for publication.

### Author contributions

Frank Tanser, Conceptualization, Data curation, Formal analysis, Supervision, Funding acquisition, Validation, Investigation, Methodology, Writing—original draft, Project administration, Writing—

review and editing; Daniel K Azongo, Investigation, Methodology, Writing—original draft, Project administration, Writing—review and editing; Alain Vandormael, Formal analysis, Writing—original draft, Writing—review and editing; Till Bärnighausen, Methodology, Writing—original draft, Writing—review and editing; Christopher Appleton, Conceptualization, Funding acquisition, Investigation, Methodology, Writing—original draft, Writing—review and editing

### Author ORCIDs
Frank Tanser  http://orcid.org/0000-0001-9797-0000
Alain Vandormael  http://orcid.org/0000-0002-5742-0511

### Ethics

Human subjects: Ethical clearance for the study was obtained from the Biomedical Research Ethics Committee of the University of KwaZulu-Natal (reference number E165/05). Prior to testing, we were required to obtain both written informed consent from parents and written assent from children participating in the survey

### Decision letter and Author response

Decision letter https://doi.org/10.7554/eLife.33065.019
Author response https://doi.org/10.7554/eLife.33065.020

## Additional files

### Supplementary files

• Supplementary file 1. Shows the adjusted prevalence ratios (aPR) for the risk factors of *Schistosoma haematobium* infection, which are parallel to the results presented in *Table 3*. Model 0 gives the univariate results and Model 1 includes all variables in the model. In Model 2, piped water coverage in the immediate community surrounding each participant has been substituted with the household-level piped water covariate.
DOI: https://doi.org/10.7554/eLife.33065.015

• Supplementary file 2. Linear probability regression models showing the impact of piped water on schistosomiasis infection in primary school children across the study area. Model 0 gives the univariate results and Model 1 gives the multivariate results for the availability of piped water in the community. Model 2 shows the instrumental variable estimation (IVE) results corresponding to Model 1, where the instrumental variable is the year that piped water was introduced into the community. Model 3 gives the multivariate results for piped water in the household and Model 4 shows the corresponding IVE results.
DOI: https://doi.org/10.7554/eLife.33065.016

• Transparent reporting form
DOI: https://doi.org/10.7554/eLife.33065.017

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
