## [Decision Letter]

Thank you for submitting your article "Impact of the scale-up of piped-water on urogenital schistosomiasis infection in rural South Africa" for consideration by *eLife*. Your article has been reviewed by three peer reviewers, and the evaluation has been overseen by Eduardo Franco as a Reviewing Editor and Prabhat Jha as the Senior Editor.

The reviewers have discussed the reviews with one another and the Reviewing Editor has drafted this decision to help you prepare a revised submission.

Summary:

You and your colleagues have conducted an impressive study that can substantially improve prevention strategies against urogenital schistosomiasis in rural South Africa. Your investigation provided compelling evidence that improvements in access to piped water could lead to reductions in disease prevalence. The design of your study and the epidemiological methods are strong features of your investigation, which bolsters the confidence in the findings. That being said, we have a few concerns that must be addressed in a revision. We summarize below the essential points that require your attention.

Essential revisions:

1) Please comment on schistosomiasis control efforts in the region, specifically school-based Mass Drug Administration (MDA) campaigns. Have these children been treated in the past, and if so, how recently compared to the cross-sectional analysis? Also important is the degree of penetration of treatment campaigns. For example, were the kids living in areas with piped water treated, and those living in areas with higher Schistosoma burdens not treated in MDA campaigns? There is ample data suggesting that MDA campaigns have imperfect penetration into communities, and often it is communities that are more accessible (perhaps those where it is easier to set up piped water) will benefit from MDA whereas those more remote communities will not. The authors must address these issues.

2) The authors assume there is little to no *S. mansoni* in the region and cite: "Appleton, J. Kvalsvig, A school-based helminth control programme successfully implemented in KwaZulu-Natal. Southern African Journal of Epidemiology and Infection 21, 55-67 (2006)." This reference does not rule out *S. mansoni* in the region. Both *S. haematobium* and *S. mansoni* have particular micro-geographies but can also be co-endemic in the same region. Does the *S. haematobium* data presented in this manuscript tell the entire story? It is plausible to assume that the regions with fewer *S. haematobium* cases are just endemic for *S. mansoni* and not *S. haematobium,* and that this was never discovered because stool examination was not performed. The authors are asked to address this as a potential weakness.

3) The statistical analysis methods must be explained in more detail. The outcome variable for the logistic regression models was positivity for urogenital schistosomiasis. Was the outcome ascertained via a combination of the results from the tests for hematuria and urine eggs?

4) You stated that the instrumental variable (IV) is "unlikely to be independently correlated with the outcome (urogenital schistosomiasis infection) at the household level". One could argue that the "complex political constellations at the district, sub-district and ward level, with rapidly changing leadership and alliances over the study period" may be associated with affluence of a given community or families. Under this assumption, those clusters of households receiving piped water may have been also exposed to better sanitation and access to drug administration campaigns. Please explain your reasons for the choice of an IV a little better so as to assuage the above concern. Explain also how you incorporated the IV in the logistic models.

5) Use two decimal places in Table 3. Consistent with modern epidemiologic thinking please delete the p value columns from Table 3 and Table 4.

6) The y axis of Figure 1 is wrong. The label indicates prevalence as a percentage but the axis is shown as a 0-1 probability scale. The overall prevalence of infection was 16.8%, which is not coherent with the graph as shown.

7) Delete "as a recent systematic review of the literature concluded" from the Abstract. The statement stands on its own, there is no need to cite the reason, which to do properly would require referencing.

8) "The most frequently infected age group is 5 to 15 years after which infection rates[…]": the term "infection rates" is rather vague and is not widely accepted in epidemiology. Please refer to incidence or prevalence, as appropriate.

9) Subsection “Derivation of community-level piped water coverage”: Why is the 2002 population-based survey not included? Households were deemed to have access to piped water if their most common source of drinking water was piped water. What is meant by most common? How was this assessed in the survey? Is this an accurate measure? How was the survey validated?

10) Please explain better the survey used to assess the level of access to piped water. You must convince readers that the survey led to sufficiently accurate estimates?

11) Results, seventh paragraph: Please mention in the Materials and methods section that you expanded on the systematic review of Grimes et al. with your own findings and explain how you did that. Keep the discussion on in the Results section.

---

## [Author Response]

Essential revisions:1) Please comment on schistosomiasis control efforts in the region, specifically school-based Mass Drug Administration (MDA) campaigns. Have these children been treated in the past, and if so, how recently compared to the cross-sectional analysis? Also important is the degree of penetration of treatment campaigns. For example, were the kids living in areas with piped water treated, and those living in areas with higher *Schistosoma* burdens not treated in MDA campaigns? There is ample data suggesting that MDA campaigns have imperfect penetration into communities, and often it is communities that are more accessible (perhaps those where it is easier to set up piped water) will benefit from MDA whereas those more remote communities will not. The authors must address these issues.

We agree that it is important to document information related to school-based treatment initiatives in the manuscript. We have added additional information to the Results (second paragraph), Discussion (sixth paragraph) and the Materials and methods (subsection “Study area”, second paragraph) sections of the manuscript to address this point. For the reasons described in these sections and summarised below, the limited ad-hoc treatment activities that did take place are unlikely to have had a bearing on our findings.

In the period leading up to the parasitological survey there were no schistosomiasis mass treatment campaigns, but the local Department of Health did undertake occasional, ad-hoc ‘test and treat’ activities in approximately 10-15% of schools in the study area at different points in time. These infrequent, schistosomiasis ‘test and treat’ activities were conducted only in schools located in the highest risk areas (where piped water coverage was lowest), and thus this could theoretically bias the result towards the null hypothesis of no piped water impact, but could not account for a spurious positive finding. However, introduction of a bias by these activities is unlikely, given the large piped-water effect sizes that we report and the fact that only 2% of participants in our survey reported receiving praziquantel in the previous 12 months (which we use as a control variable in the statistical model). In any event, if such a ‘residual confounding’ effect did exist (because of the effect of a previous treatment initiative, for example), it would be completely controlled for in the instrumental variable approach utilised in the paper, the results of which confirmed all of our findings reported in the manuscript.

2) The authors assume there is little to no S. mansoni in the region and cite: "Appleton, J. Kvalsvig, A school-based helminth control programme successfully implemented in KwaZulu-Natal. Southern African Journal of Epidemiology and Infection 21, 55-67 (2006)." This reference does not rule out S. mansoni in the region. Both S. haematobium and S. mansoni have particular micro-geographies but can also be co-endemic in the same region. Does the S. haematobium data presented in this manuscript tell the entire story? It is plausible to assume that the regions with fewer S. haematobium cases are just endemic for S. mansoni and not S. haematobium, and that this was never discovered because stool examination was not performed. The authors are asked to address this as a potential weakness.

We agree that additional information is required in the manuscript regarding the distribution of *Schistosoma mansoni*. We have now included additional information to this effect including a key reference (Appleton and Madsen, 2012) which outlines in detail the reason for *Schistosoma mansoni* being largely absent in the coastal plain of north-eastern KwaZulu-Natal, where the study was conducted (end of subsection “Schistosomiasis in South Africa”).

3) The statistical analysis methods must be explained in more detail. The outcome variable for the logistic regression models was positivity for urogenital schistosomiasis. Was the outcome ascertained via a combination of the results from the tests for hematuria and urine eggs?

In response to this comment,we have considerably expanded the subsection “Statistical analysis” and in additionhave added a sentence to describe the fact that children were considered to be infected with *Schistosoma haematobium* on the basis of the microscopy result.

4) You stated that the instrumental variable (IV) is "unlikely to be independently correlated with the outcome (urogenital schistosomiasis infection) at the household level". One could argue that the "complex political constellations at the district, sub-district and ward level, with rapidly changing leadership and alliances over the study period" may be associated with affluence of a given community or families. Under this assumption, those clusters of households receiving piped water may have been also exposed to better sanitation and access to drug administration campaigns. Please explain your reasons for the choice of an IV a little better so as to assuage the above concern. Explain also how you incorporated the IV in the logistic models.

We agree with this reviewer’s comment and as requested have further expanded and strengthened this section (subsection “Adjustment for unobserved social and environmental predictors of infection”) to describe the rapidly-changing multitude of political, social, and structural and factors leading to delivery of piped water in a particular community at a particular point in time. The joint presence of these constantly changing factors leading to piped water delivery at a specific time-point was likely as-good-as random regarding other potential determinants of infection, such as wealth or coverage with health services. It is thus unlikely to be independently correlated with the outcome (urogenital schistosomiasis infection) and likely meets the exclusion restriction.

Furthermore, we have explicitly controlled for a wide range of environmental and socio-demographic factors including all of the sources of potential confounding specifically mentioned here by the reviewer – i.e. variations in “affluence” (household assets), “sanitation” (presence of toilet in the household) and “exposure to drug administration campaigns” (praziquantel in the last 12 months) – both in our baseline statistical models (Table 3,Table 4) as well as in our instrumental variable regression analysis (Supplementary file 2). In our response under point 1 (above), we describe in detail why treatment activities (that occurred only in the highest risk areas) are unlikely to have resulted in a bias towards the null hypothesis given the large piped-water effect sizes that we report and the fact that only 2% of participants in our survey reported receiving praziquantel in the previous 12 months.

We have also added additional explanation and an appropriate citation regarding on the IV statistical analysis (subsection “Adjustment for unobserved social and environmental predictors of infection”).

5) Use two decimal places in Table 3. Consistent with modern epidemiologic thinking please delete the p value columns from Table 3 and Table 4.

Table 3 now shows the results to two decimal places.

In respect of the inclusion of p-values, we understand the debate and are aware of their potential misuse in science. However, we believe there is value to the inclusion of both confidence intervals and p-values in the manuscript. In response to this comment we have now systematically added confidence intervals to all our effect size estimates reported in the paper as well as in the tables and have retained the p-values.

Further, many colleagues and readers of the article who could cite our paper will utilise both the confidence intervals and p-values in making their assessment of the strength of the findings. For example, a recent article published in *eLife* on the impact of water sanitation on health outcomes uses p-values in both the text and tables:

Audrie Lin, Benjamin F Arnold, Andrew N Mertens, et al. 2017. Effects of water, sanitation, handwashing, and nutritional interventions on telomere length among children in a cluster-randomized controlled trial in rural Bangladesh. *eLife* 6: e29365.

https://elifesciences.org/articles/29365

6) The y axis of Figure 1 is wrong. The label indicates prevalence as a percentage but the axis is shown as a 0-1 probability scale. The overall prevalence of infection was 16.8%, which is not coherent with the graph as shown.Thank you for pointing this out. This has been corrected and the caption clarified (Figure 1).7) Delete "as a recent systematic review of the literature concluded" from the Abstract. The statement stands on its own, there is no need to cite the reason, which to do properly would require referencing.We agree; and have removed this phrase from the Abstract.8) "The most frequently infected age group is 5 to 15 years after which infection rates[…]": the term "infection rates" is rather vague and is not widely accepted in epidemiology. Please refer to incidence or prevalence, as appropriate.

We agree; and have replaced the term “infection rates” with “prevalence” (subsection “Schistosomiasis in South Africa”).

9) Subsection “Derivation of community-level piped water coverage”: Why is the 2002 population-based survey not included? Households were deemed to have access to piped water if their most common source of drinking water was piped water. What is meant by most common? How was this assessed in the survey? Is this an accurate measure? How was the survey validated?

We agree that it is important to strengthen and expand this section, which we have now done (subsection “Derivation of community-level piped-water coverage”).

The household socio-economic survey (which includes piped-water access) is an add-on component of the demographic core survey that is not conducted every year. Thus in 2002 this add-on survey did not take place. We therefore used all five available population-based surveys (2001, 2003/2004, 2005, 2006, 2007) conducted up to the year of parasitological survey (2007) in the surveillance area to measure coverage of piped-water. In this section, we have now added a reference to an article that describes the household socio-economic surveys in detail (Tanser et al., 2008).

10) Please explain better the survey used to assess the level of access to piped water. You must convince readers that the survey led to sufficiently accurate estimates?

Further to the additions noted under point 9, we have provided further detailed explanation of the piped water survey (subsection “Derivation of community-level piped-water coverage”). All households in the entire surveillance area were asked a detailed question on “which is the most often used source of drinking-water in this household?” The household key informant was asked to select the ‘most common’ source from a list of 11 different sources of water listed.

In addition, we have added information (Results, fourth paragraph) to reflect the fact that there was a high degree of temporal and spatial coherence and plausibility in the patterns of expansion of piped water access across the five respective surveys, providing further assurance of the quality of the underlying population-based data.11) Results, seventh paragraph: Please mention in the Materials and methods section that you expanded on the systematic review of Grimes et al. with your own findings and explain how you did that. Keep the discussion on in the Results section.

We have added a section to the Materials and methods section (subsection “Statistical analysis”, last paragraph) detailing the methods used for this meta-analysis.